# Regenerative Anterior Cruciate Ligament Healing in Youth and Adolescent Athletes: The Emerging Age of Recovery Science

**DOI:** 10.3390/jfmk9020080

**Published:** 2024-04-23

**Authors:** John Nyland, Michael N. Sirignano, Jarod Richards, Ryan J. Krupp

**Affiliations:** Norton Orthopedic Institute, 9880 Angie’s Way, Suite 250, Louisville, KY 40241, USAjarod.richards@louisville.edu (J.R.); ryan.krupp@nortonhealthcare.org (R.J.K.)

**Keywords:** regenerative medicine, orthobiologics, injury prevention, recovery science, exercise

## Abstract

Anterior cruciate ligament (ACL) injuries mainly arise from non-contact mechanisms during sport performance, with most injuries occurring among youth or adolescent-age athletes, particularly females. The growing popularity of elite-level sport training has increased the total volume, intensity and frequency of exercise and competition loading to levels that may exceed natural healing capacity. Growing evidence suggests that the prevailing mechanism that leads to non-contact ACL injury from sudden mechanical fatigue failure may be accumulated microtrauma. Given the consequences of primary ACL injury on the future health and quality of life of youth and adolescent athletes, the objective of this review is to identify key “recovery science” factors that can help prevent these injuries. Recovery science is any aspect of sports training (type, volume, intensity, frequency), nutrition, and sleep/rest or other therapeutic modalities that may prevent the accumulated microtrauma that precedes non-contact ACL injury from sudden mechanical fatigue failure. This review discusses ACL injury epidemiology, current surgical efficacy, the native ACL vascular network, regional ACL histological complexities such as the entheses and crimp patterns, extracellular matrix remodeling, the concept of causal histogenesis, exercise dosage and ligament metabolism, central nervous system reorganization post-ACL rupture, homeostasis regulation, nutrition, sleep and the autonomic nervous system. Based on this information, now may be a good time to re-think primary ACL injury prevention strategies with greater use of modified sport training, improved active recovery that includes well-planned nutrition, and healthy sleep patterns. The scientific rationale behind the efficacy of regenerative orthobiologics and concomitant therapies for primary ACL injury prevention in youth and adolescent athletes are also discussed.

## 1. Introduction

When one thinks about an anterior cruciate ligament (ACL) injury, they often picture a vicious on-field collision between two competitors. However, 70% of these injuries are associated with non-contact mechanisms [1], and many of these injuries may be associated with repetitive, submaximal loading [2,3]. Through reconstructive surgery and rehabilitation, most athletes who sustain an ACL rupture successfully return back to their sport, but with less than ideal outcomes or performance levels [4,5]. Among athletically active individuals, ACL injuries have an incidence of 75/100,000 person-years [6]. Under the premise that the ACL and menisci of youth and adolescent athletes likely possess a better healing capacity following musculoskeletal injury than that of older individuals [7,8], we describe a recovery science plan to decrease the high non-contact primary ACL injury incidence among this uniquely vulnerable and under-served population [9,10] who regularly participate in sport training frameworks where guidelines, rules, and regulations have been established with minimal scientific evidence [11]. With appropriate nutrient delivery and exercise prescription, the native neurovascular system may be best capable of delivering the ideal blend, dosage, and delivery timing of biochemical healing agents to break the negative cycle of mechanical fatigue-related sudden ACL failure. Growing evidence suggests that the prevailing mechanism that leads to non-contact ACL injury from sudden mechanical fatigue failure is accumulated microtrauma. Given the consequences of primary ACL injury on the future health and quality of life of youth and adolescent athletes, the objective of this review is to identify key “recovery science” factors that can help prevent these injuries. Recovery science is any aspect of sports training (type, volume, intensity, frequency), nutrition, and sleep/rest or other therapeutic modalities that may prevent the accumulated microtrauma that precedes non-contact ACL injury from sudden mechanical fatigue failure. This review discusses ACL injury epidemiology, current surgical efficacy, the native ACL vascular network, regional ACL histological complexities such as the entheses and crimp patterns, extracellular matrix remodeling, the concept of causal histogenesis, exercise dosage and ligament metabolism, central nervous system reorganization post-ACL rupture, homeostasis regulation, nutrition, sleep and the autonomic nervous system. This review calls for re-thinking primary ACL injury prevention strategies with greater use of modified sport training, improved active recovery that includes well-planned nutrition, and healthy sleep patterns. The scientific rationale behind the efficacy of regenerative orthobiologics and concomitant therapies for primary ACL injury prevention in youth and adolescent athletes is also discussed.

## 2. Epidemiology and Current Interventions

Youth and adolescent organized sports participation continues to grow in Western countries [12], with children starting sports at younger ages and competing at higher levels [13]. Overuse injuries of the musculoskeletal system are prevalent in the USA with more than one-third involving ligaments [14]. From 2005 to 2015 in Victoria, Australia, among 5–14-year-old athletes, the number of hospital-treated ACL injuries increased by 147.8% [15]. This dramatic increase suggests a growing public health crisis as many of these athletes will eventually be more likely to develop knee osteoarthritis [16,17]. Youth and adolescent athletes represent the population most at risk for non-contact ACL injuries, particularly athletically active girls [1,18].

As more youth and adolescent athletes become involved in elite-type sport training, with greater concentration on a single sport with higher year-round training volume, injury rates increase [19], and less time is available for free, unstructured play [20]. Growing basic [2,21,22,23,24,25,26] and clinical science [3] evidence suggests that sudden ACL failure from non-contact mechanisms in this population represents accumulated mechanical fatigue-failure from the overuse [2,27,28] associated with molecular collagen unfolding and cross-link weakening [26,29]. Although pre-injury and post-surgical neuromuscular training has helped decrease primary ACL injury and re-injury incidence, there may be an additional path available to significantly decrease their incidence in a manner that better appreciates the growing awareness and understanding of this somewhat silent injury mechanism with better primary injury prevention strategies [2,15,27] and improved healthcare policies [30].

## 3. The Limitations of ACL Reconstruction and Rehabilitation

The differences between native ACL neurosensory characteristics and the lack thereof in the replacement graft are enormous [31,32,33]. Because natural ACL neurosensory function is not effectively restored by ACL reconstruction, to enable continued high-level sports function, neuroplastic central and peripheral nervous system adaptations must occur to create compensatory neuromuscular activation patterns [32,33]. The replacement graft also possesses considerably different morphology and dimensions than the native ACL, primarily solely serving as a high-density collagenous tissue scaffold for long-term ligamentization remodeling. The term “anatomical” ACL reconstruction only refers to graft placement and fixation somewhere within the native ACL insertional sites or “footprints”. Otherwise, nothing in association with graft implantation is anatomically relevant to the native ACL in any way. In fact, to insure “anatomical” graft placement and fixation, the surgeon generally has to completely remove the most neurovascularly rich ruptured ACL region, its femoral insertion, which may possess a natural healing capacity similar to the medial collateral ligament [22,34,35].

Even in a best case scenario with a strong graft anatomically implanted with strong fixation, the remodeling or “ligamentization” process takes a long and somewhat unpredictable amount of time, with highly variable levels of completeness, and a lack of pre-morbid neurosensory function restoration [31,32,33]. Although anatomical graft placement simulates more natural knee kinematics, subtle tibiofemoral joint loading location and pressure differences remain [30], and graft fixation either through intra-tunnel or socket screws or through extra-cortical buttons or washer-post combinations does not transfer loading forces in a manner that physiologically replicates native ACL function [36,37,38]. Current reconstruction methods use homogenous grafts with considerably different morphology, histology, and natural function compared to the native ACL. Regardless of which graft type is used, patient return to competitive sport outcomes are less than ideal [4,5,31]. Much remains to be learned about reconstructing a ruptured ACL in a manner that actually restores its complex three-dimensional anatomy, physiology, and regional structural heterogeneity [39,40,41]. In current practice, the likelihood of an ACL rupture progressing to knee osteoarthritis is similar regardless of whether it is treated non-surgically or with reparative or reconstructive surgery [42]. In fact, many of these patients will experience some level of degenerative disease over time, often with a disabling outcome that robs many of their exercise or sports potential early in life, and raising their risk of heart disease, diabetes mellitus, and obesity [31]. From this perspective, the term “anatomic” ACL reconstruction is a misnomer. In particular, orthopedic surgery, although adequately restoring essential biomechanical properties remains perplexed as how to restore pre-morbid natural neurosensorimotor function through free graft use [31].

Another ACL reconstruction concern is the potential supplementary peripheral knee joint stabilization need for athletes who regularly participate in cutting or pivoting sports such as soccer, basketball or American football [43]. These procedures have gained popularity for revision ACL reconstruction cases or for primary procedures in patients deemed at “high re-tear risk”, which may be a synonym for many adolescent athletes. The Stability I trial reported a decreased re-injury rate following ACL reconstruction with a hamstring tendon autograft when a supplemental lateral extra-articular tenodesis was performed, compared to ACL reconstruction alone [44]. The Stability II trial comparing quadriceps tendon ACL reconstruction and supplemental lateral extra-articular tenodesis is in progress [44]. However, concerns related to complication profiles for these combined surgical procedures are still evolving [43,45]. The Stability I study found increased knee pain in the combined surgical procedure group over the initial 6 months post-surgery, and a prior study reported increased post-surgical knee stiffness [44]. Implications for rehabilitation protocols and return to play criteria have yet to be elucidated [46].

## 4. Non-Surgical and Surgical ACL Repair: Growing Evidence

A detriment to ruptured ACL healing is the lack of any tissue that bridges the gap between the torn ends [47]. By placing the knee in a brace at 90° flexion for 4 weeks following an acute ACL injury with a tear gap distance < 4–6 mm and with an intact synovial envelope, the Cross Bracing Protocol represents a non-surgical method that can help the torn ACL heal [47]. After 4 weeks of immobilization, knee range of motion is progressively increased as are therapeutic exercises that target lower limb neuromuscular control, muscle strengthening power, and functional training to enable return to sport and recreational activities [47]. Complete cessation of brace use occurs at 12 weeks. Among 80 consecutive patients with a mean age of 26 ± 10 years of age (range = 10 to 58 years of age), 90% displayed MRI evidence of ACL healing at 3 months with a grade I ACL Osteoarthritis Score [47].

Four different ACL surgical repair procedures have also come to the forefront: 1. bridge-enhanced ACL repair (BEAR), 2. proximal anchor fixation with suture anchors, 3. dynamic intra-ligamentary stabilization (DIS), and 4. internal bracing [48]. To facilitate healing, the BEAR procedure combines proximal ACL rupture refixation with the implantation of a collagen sponge soaked in autologous blood, and internal ligament brace supplementation. In DIS, to approximate ACL repair construct stiffness that better matches that of the normal ACL, an internal brace augmentation cord attached to the injured ACL is connected to a spring that is anchored to a metal screw. In general, each of these methods is more effective when used with patients who have experienced more proximal, acute ruptures. The ideal indication for primary ACL repair remains unclear, and although early results are promising [49], long-term outcome efficacy studies are ongoing. In the presence of sub-optimal outcomes post-ACL reconstruction, and limited evidence regarding non-surgical or surgical ACL repair, perhaps greater attention should be placed on improving our understanding of factors that in combination may positively influence primary ACL healing following exercise- or sports training-induced microtrauma [2,27,31].

## 5. Native ACL Blood Supply

Most of the blood supply to the ACL is provided by the middle geniculate artery [50], and the distal third is vascularized by a peri-ligamentous network from terminal branches of the middle and inferior geniculate arteries [51] (Figure 1). The ACL is nearly entirely supplied by branches traveling through the synovium, rather than from the bony insertions. Approximately 0.5 cm proximal to the tibial insertion, blood vessels are absent where contact occurs with the intercondylar femoral notch when the knee is fully extended [51]. From this “physiological impingement”, “chondrocyte-like” cells that form Type II collagen exist in alignment with longitudinal (Type I) collagen fibers [52].

The middle and inferior geniculate arteries provide branches to the synovial tissue that covers the ACL (Figure 2 and Figure 3). From the synovial tissue, small blood vessels penetrate the ACL centripetally and anastomose with longitudinally oriented intraligamentous vessels. The blood vessel volume within the ACL substance is considerably less than that of the synovial tissue. Blood vessels within the ACL substance occur in the loose connective tissue between longitudinal collagen fiber bundles, which protects them from shear forces. The richest ACL blood supply resides within the proximal ACL [53].

The ACL insertion osseous junctions consist of fibrocartilage that does not contribute to its vascularization. These regions help modulate the modulus of elasticity difference as the ACL transitions from soft tissue to bone, also serving as “stretching brakes” to reduce horizontal ligament shortening. This absence of blood vessels is likely related to the repetitious compressive strains that occur in this region [51]. Nourishment to avascular ACL tissue regions occurs through interstitial fluid convection, particularly at the distal ACL region. Intermittent compressive strain in avascular regions facilitates nutrition via convection, similar to how articular cartilage receives nutrition. Lymphatics also collect and remove metabolic waste fluids and solutes from the interstitial space. In addition to facilitating nutrition exchange, water inside the ACL serves a biomechanical purpose. When ACL tensile forces increase, interstitial fluids shift to the loose connective tissues located between collagen fiber bundles. The limited blood and lymph supply in the avascular distal ACL explains the poor natural healing responses that exist in this region. Synovial fluid diffusion can support ACL collagen synthesis [54]. The ACL is supplied by sensory afferent and vasomotor efferent nerve fibers. Vasomotor efferent nerve fibers are part of the sympathetic nervous system serving a mainly vasoconstrictor function, that with dysregulation may have an additional role in articular disease pathogenesis [31,55]. Autoimmune system dysregulation associated with heightened sympathetic nervous system activation may be directly related to knee osteoarthritis development [56].

## 6. Regenerative Orthobiologics

A growing sports medicine area is regenerative orthopedics or orthobiologics. Through cellular therapies, the biological healing response may be enhanced for many orthopedic conditions. Traditionally, ligaments were considered mechanical bands that did not respond to exercise; however, it is now clearly understood that they do [3,31,57]. Differing combinations of stem cells, plasma rich in platelets, exosomes (extracellular vesicles) and other biological nutrients may improve tissue healing outcomes [58]. In combination with these biochemical factors, physical therapeutic agents such as aerobic exercise (or hyperbaric oxygen), heat, cold, sound, shockwaves, light, or electromagnetic fields may further enhance the healing timetable [59]. By restoring tissue healing homeostasis [3,27,31], in the microtraumatic environment, regenerative processes have the potential to completely replicate the morphology, histology, physiology and mechanical property complexities of the entire natural ACL.

The keys to regenerative ACL therapies include the cell type used, dosage, delivery timing, application frequency, and what, if any, concomitant physical therapeutic agents are used. Our current level of understanding in this area is weak [31]. Few athletes can identify a particular knee symptom such as pain, effusion, stiffness, laxity, or dynamic instability during the accumulative fatigue-related overuse ligamentopathy that leads to sudden, non-contact ACL failure. However, while in this condition, they are more likely to experience the sudden “pop” that they remember for the rest of their life when they rupture their ACL when simply warming up, practicing, or more often, competing in their sport. Improved primary prevention strategies present several inherent advantages for reducing the accumulated microtrauma that culminates in sudden ACL rupture [3,27,31]. Most importantly, the native ACL macro-structure is still intact, including its synovial sheath, and its neurovascular network. In the presence of better designed active recovery, modified training, proper nutrition, and healthy sleep patterns, a “best case scenario” may be created that enables natural ACL healing and remodeling to generate a stronger tissue.

## 7. Regional ACL Histological Differences and Complexities

To improve ACL surgical repair or reconstruction efficacy, tissue engineers have been attempting to better replicate zones of differing ACL tissue heterogeneity. Despite these advances, one common acknowledgment in these reports is that we have a long way to go [39,40,41]. The native crimp pattern histological zones immediately adjacent to the entheses (Figure 4) provide natural braking that dissipates acute loading forces through both the anteromedial and posterolateral ACL components (Figure 5). As little is known about the histology, morphology and physiology of these histologically different ACL regions, even less is known about how they could become stronger through the modified sports training that enables better tissue healing and remodeling [3,27,31]. What is known is based primarily on biological and biomechanical comparative animal model studies [26,29,60] and in vitro cadaveric biomechanical studies [2,21,22,23,25,35]. Comparative animal model studies strongly suggest that the positive effects of regular exercise training is directly related to the ability of the circulatory system to deliver biological healing agents directly to the ACL, or via synovial fluid diffusion. Youth and adolescent athletes likely possess a more robust knee joint healing capacity than adults, further supporting this concept [7,8].

## 8. The Extracellular Matrix

A healthy ACL is dependent on a collagen-rich extracellular matrix (ECM), which derives its structure and function from longitudinally aligned type I collagen and type III collagen crosslinking in combination with interstitial water and minerals. In a weakened or diseased state, or with nutritional insufficiency, a deficient ECM may not be able to withstand the mechanical demands of normal activity. While poor nutrition, genetics, and disease can predispose the ACL to rupture from microtrauma accumulation, adequate nutrition combined with appropriate exercise dosage and proper inter-training session recovery may improve ECM structure and function by better restoring homeostasis [2,27,31]. Through its abundant peripheral circulatory system volume, muscle tissue has an approximately 7.5 times more robust healing capacity than ligament [52], and the lower metabolic rate in a ligament is more likely to result in slower, less complete healing after injury [62,63,64]. Within the native ACL, histologically distinct zones of relatively good or poor circulatory system regions exist for delivering healing biological factors to micro-damaged tissues [35,50,51,52,53]. With repetitive loading, the ACL undergoes varying microtraumatic damage levels throughout the predominantly type I collagen of the ECM [2,21,22,23]. With sufficient recovery time and activities that promote active recovery, the native ACL can become stronger, remodeling through the development of a greater concentration of well-aligned longitudinal type I collagen fibers and type III collagen insertion area crosslinks [2,27]. In the presence of repetitious loading in combination with inadequate recovery, however, accumulated microtrauma may create ACL collagen degradation “ligamentopathy” that predisposes it to sudden rupture [2,3], even though it appears completely healthy with clinical examination and under direct macroscopic evaluation using an arthroscope. A diet consisting of low-nutrient-density foods and unhealthy sleep patterns may further exacerbate ACL degradation.

To better preserve native ACL sensorimotor and neurovascular properties, a growing evidence base supports repair rather than reconstruction when feasible [47,49], particularly in younger patients who have sustained acute femoral insertional lesions. The pioneers of these procedures [54] have acknowledged the superior healing capacity of younger patients when an intra-lesion scaffold is used. Early outcome reports for contemporary ACL repair procedures have been encouraging [47,49,65,66,67], but only time and well-designed long-term studies will confirm procedural efficacy for restoring premorbid mechanical, kinematic and neurosensorimotor function.

By better restoring homeostasis post-microtrauma [2,3], however, natural ACL anatomy, physiology, biomechanics and essential neurosensorimotor function may be completely restored, or even enhanced with complete collagen crimp pattern replication, and complete enthesis preservation without the need for surgery. A previously untapped opportunity may exist to better prevent accumulated microtrauma, overuse related sudden ACL fatigue failure from non-contact mechanisms among youth and adolescent athletes, that takes full advantage of their more robust peripheral neurovascular networks.

## 9. Causal Tissue Histogenesis

Complex dynamic cyclic compression, tension and shear forces influence both ACL physiologic and region-specific tissue neurovascularity [68]. This neurovascularity may change with age, disease, or injury (altering the local microvascular environment). Articular cartilage is avascular, with nutrition arising primarily from synovial fluid diffusion throughout its matrix. Knee movement and weightbearing compression actively pumps nutrient solutes into, and metabolic by-products out of the articular cartilage matrix. Cyclic compressive knee joint loads increase both synovial fluid production and the rate of larger solute (growth factors, hormones, enzymes, and cytokine) distribution across and beneath the articular cartilage surface, enhancing chondrocyte metabolism [69]. In the absence of sufficient knee joint movement and loading, inadequate nutrient transport leads to accumulated pools of stagnant, nutrient depleted synovial fluid acidic wastes such as lactate and carbon dioxide accumulation, creating unhealthy articular cartilage [69].

Although the middle genicular artery provides primary ACL blood supply, it is also supplied by the accessory middle genicular artery, posterior capsular and inferior genicular arteries [50,51]. Regardless of specimen age, vascular network connections exist between the synovial membrane covering Hoffa’s fat pad, the ACL and the anterior meniscal horns. The ACL derives most of its extrinsic vasculature from the intracapsular horizontal tract of the middle genicular artery, with larger vessels descending primarily along the ACL surface that faces the PCL. The ACL is also supplied by collateral branches of the arteries directed to the roof of the intercondylar notch and the lateral femoral condyle. Infrapatellar ramifications of the inferior genicular artery provide a lesser vascular supply to the distal ACL [50]. For both cruciate ligaments, the intrinsic blood supply originates primarily from superficial, mostly sub-synovial vessels, which penetrate the ligament body obliquely or transversally at varying levels and depths [50]. After a short endoligamentous course, these vessels (arterioles, pre-capillaries, and capillaries) located in the interfascicular septa with venules and a few thin nerve fibers split longitudinally upward or downward parallel to the collagen fiber bundles.

Since the ACL vascular network can deliver essential biological factors, its limited healing capacity following partial or complete rupture is likely related more to anatomical or mechanical factors [70], where an adequate provisional bridge cannot form between its torn ends [71]. There is a direct correlation between ACL tensile strength and decreased injury incidence and severity [72]. With the increasing popularity of the high-intensity, high-frequency and high-total-volume elite training and competition model for youth and adolescent athletes, the healthcare team should be highly aware of all factors that may combine to weaken the ACL ECM [2,3,27,31].

## 10. Exercise Dosage

Murine model studies involving intermittent aerobic exercise for 3 weeks or more are known to strengthen knee ligaments more than spontaneous or confined movement activities [60,73,74,75,76,77,78,79], also increasing brain-derived neurotrophic factor and hippocampal neurogenesis [80]. Cabaud et al. [60] studied the effects of different aerobic endurance exercise frequency and duration combinations on the ACL mechanical properties of rats that participated in varying exercise programs for a 10 week duration. In all exercise conditions, the ACL became stronger compared to no exercise; however, the most consistent ACL strength and stiffness improvements occurred in the high-frequency, low-duration exercise condition groups. Use of an aerobic endurance exercise program that simulated significant aerobic exercise in humans (i.e., VO_2_ maximum of >80%) made the ACL “less elastic” and more resistant to deformation [60]. The authors concluded that: 1. endurance-type exercise in general had a positive effect on ACL strength and stiffness; 2. the greatest increase in ACL strength and stiffness occurred after high-frequency, low-duration exercise, and the least significant changes occurred from low-frequency, high-duration exercise; and 3. longer duration or less frequent exercise may decrease ACL strength and stiffness [60]. Both murine model [29] and clinical studies [3] have confirmed ACL weakening across repetitive, submaximal intensity exercise sessions.

Shorter, more acute high-impact exercise sessions are also known to increase collagen synthesis in bone, particularly when supplemented with proper nutrition [81]. Vitamin C is not just vital to collagen synthesis, but it also activates collagen-crosslinking enzymes lysyl oxidase, prolyl and lysyl hydroxylases [81]. Increased ECM crosslinking improves its mechanical properties even in the absence of any total collagen volume increase. A single 10 min bout of cyclic stretch followed by 6 h of rest increased the collagen content of engineered ligament grafts more than a continuous stretch program [82]. As few as 10 maximal effort vertical jumps/day performed 3 days/week increased the bone mineral density of female college students [83]. This suggests that ligaments get a maximal collagen synthesis stimulus from short periods of activity and comparatively longer intervening rest periods. Gelatin supplementation augments collagen synthesis post-exercise. An accelerated collagen synthesis rate begins as early as 4 h after an initial exercise bout (5 h post-gelatin supplementation) and is maintained for 72 h post-exercise [81]. Appropriately timed gelatin and vitamin C ingestion in association with properly dosed intermittent exercise could benefit ACL tissue repair and injury prevention [81,83].

In a rat model study, low-duration (30 min per day), high-frequency (6 days/week) training was found to be the most beneficial training regimen to increase ACL strength and stiffness; however, groups that trained only 3 days/week displayed similar ACL strength as the higher frequency exercise groups [60]. Cabaud et al. [60] summarized that a combination of increased exercise training frequency and decreased duration produced the most significant ACL strength and stiffness increases, and regular exercise was more beneficial than irregular or sporadic exercises. The proliferation of fibroblast collagen synthesis is closely related to oxygen availability [84,85] (Figure 6).

## 11. Traumatic ACL Rupture and CNS Reorganizational Plasticity

Sudden ACL rupture modifies ascending afferent CNS pathways and signals, creating a deafferentation injury that promulgates cortical re-organization [31,86]. This deafferentation cannot be reversed from other knee joint somatosensory inputs or by the mechanical stability provided by ACL reconstruction [31,86]. Cortical neuroplasticity enables the CNS to adapt to the different peripheral stimulus that it receives from the ascending afferent pathways with limited or no input coming from the injured ACL [32]. Since the native ACL has abundant mechanoreceptors, particularly near its proximal insertion, it is important to attempt to preserve or to re-establish any and all physiological connections capable of inciting CNS organization [79]. Patients with an ACL rupture display higher visual cortex activation to compensate for lost ACL proprioceptive input [31,87]. Collagen synthesis and fibroblast proliferation are closely related to oxygen availability [84,85,88]. Both acute and regular aerobic exercise influence ACL tissue homeostasis and adaptation through enhanced oxygen uptake and improved modulation of associated reactive oxygen species [88].

## 12. Homeostasis Modulation

Irregular, poorly planned mechanical loading can cause pathological changes from an imbalance between ECM degradation and synthesis [2,3,31,89]. In addition to overuse, abnormal loads associated with repetitive or excessive disuse, compression or shear forces can create a pathologic response [3]. When the ACL undergoes high dynamic loads, fatigue damage accumulates in combination with altered crimp properties. Repetitious, sub-maximal ACL loading degrades the ECM [2,3]. Adjusted loading frequency and duration can improve ACL ECM mechanical properties with increased collagen synthesis with a shift from more catabolic to more anabolic molecular mechanisms [90].

Homeostasis modulation represents a dynamic system with both biological and functional/actual endpoints being dependent upon ACL tissue adaptation. The biological endpoint is the point where the ACL ruptures. The functional endpoint represents an individual’s exercise tolerance level, which is naturally below the biological endpoint [88]. The distance between biological and functional endpoints represents the zone that should be targeted to induce the positive adaptations needed to extend the functional endpoint (ACL failure resistance) [88]. A key adaptive factor is intermittent exercise [2,3]. During normal recovery, any previous exercise-induced reactive oxygen species production (i.e., free radicals) is compensated for by an upregulated antioxidant system. But with chronic exercise exposure from overtraining, high reactive oxygen species levels can debilitate the antioxidant system impairing cell function, increasing macromolecular damage, increasing apoptosis, and promoting tissue necrosis [88]. Antioxidants counteract the harmful effects of reactive oxygen species [88]. When applied with the correct dose, exercise can serve an important pre-conditioning function creating an appropriate level of inflammation that serves as a protective process for ECM healing and damage repair [88]. During exercise recovery, the body adapts to any metabolic and oxidative system stresses. During these periods, in addition to storing more glycogen in the skeletal muscle to be better prepared for future performance, the body also upregulates its antioxidant and oxidative tissue damage repair systems [88]. At a given exercise intensity, individuals involved in regular, well-planned exercise programs recover with higher mitochondria levels, and lower reactive oxygen species levels compared to untrained individuals.

Therefore, although excessive physical exercise is detrimental to tissue healing [2,3], well-planned progressive training enables ACL cells to more easily detoxify reactive oxygen species, partly through antioxidant system activity upregulation. Severe or more intense exercise regimens increase accumulated oxidative damage, even among healthy subjects [88]. Poorly trained, or poorly nourished athletes are at an even greater risk for sustaining oxidative stress [88]. The oxidative damage caused by a single acute exercise session does not likely decrease tissue viability risking oxidative damage, but rather, with proper regulation it provides a vital adaptation factor that facilitates cell membrane and protein remodeling [88]. Exercise studies suggest that moderate oxidative stress accumulation levels that promote a slight increase in specific oxidative damage markers are needed to activate the adaptive responses that lead to enhanced physiological function.

High oxygen consumption with endurance running has a strong relationship with metabolic system function and the development of dynamic multi-functional oxygen reduction reaction (redox) homeostasis. In contrast, the increasing physical inactivity associated with modern lifestyles may suppress these adaptive capacities, both in metabolic and redox homeostasis, leading to increased reactive oxygen specific-related disease [88]. Homeostasis redox signaling was a vital function of human evolution with regular aerobic exercise increasing antioxidant and oxidative damage repair system efficiency. The moderate oxidative damage caused from a single exercise bout is crucial for cellular membrane remodeling, protein turnover, transcriptional cellular regulation and a wide range of metabolic and redox processes to occur [88]. The accumulated overuse associated with elite-level training, however, without sufficient recovery, may create tissue specific degenerative effects [2,3,27,31].

Exercise load characteristics can influence both tissue histology and the synovial fluid environment. Pauwels [68] described the “causal histogenesis” theory which suggested that compressive forces on mesenchymal stem cells created by collagen fiber tension created fibroblast transformation to chondroid cells. Cruciate ligament compressive forces can stimulate region-specific chondrogenesis [91]. Knee osteoarthritis is stimulated by a chronic inflammatory environment within the knee, particularly through the production of proinflammatory cytokines [91,92].

When tissues are perfused with blood during exercise, essential amino acid availability can increase ACL collagen synthesis [93]. Like muscle, the ACL responds to mechanical loads with hypertrophy from more densely packed and stronger collagen fibrils which can better protect the knee joint from injury [93]. Growing evidence supports the benefits associated with well-timed collagen peptide ingestion and amino acid bioavailability needed to increase ACL collagen synthesis, reduce knee pain, and prevent or delay osteoarthritis [57,74,94]. Through increased hormonal release, exercise is a potent collagen turnover regulator that upregulates collagen synthesis [93]. Amino acids serve a highly supportive role in ACL collagen synthesis.

## 13. Nutrition and Exercise

Given that ACL circulation is compromised with knee joint loading, it is sensible to increase circulating amino acid bioavailability levels before exercise [95]. A major ECM adaptation stimulus is strain, which prompts fibroblasts to increase collagen synthesis [95]. The timing of amino acid ingestion should be optimized to ensure peak nutrient bioavailability as blood flow to the injury site is enhanced [96]. Maintaining adequate nutrient bioavailability to support strain-induced ACL ECM collagen synthesis is important [96].

Consumption of a gelatin- and vitamin C-rich supplement increases the human serum bioavailability of the amino acid components that form collagen [81]. By 1 h following gelatin supplement consumption, human serum enriched with amino acids increase engineered ligament collagen content and mechanical properties. Subjects who consumed high amounts of gelatin 1 h before 6 min of jump roping showed collagen synthesis levels twice that of either placebo or low-gelatin concentration dose comparison groups [81]. Gelatin supplementation has previously been shown to improve connective tissue structure and function [97]. McAlindon et al. [97] reported that consumption of 10 g collagen hydrolysate/day resulted in increased knee collagen levels. In agreement with this finding, a 24-week randomized clinical study of athletes showed that collagen hydrolysate significantly decreased knee pain. Musculoskeletal collagen synthesis was greater with gelatin consumption compared to consumption of specific amino acids with glycine, proline, hydroyproline, and hydroxylysine blood levels peaking by 1 h after ingestion [97]. In contrast, lower enrichment (lysine) amino acid consumption or consumption of less gelatin resulted in peak blood levels 30 min following ingestion. This agrees with previous work that showed that peak blood hydroxyproline levels (collagen precursor) occurred later as the gelatin dose increased [81]. This supports the hypothesis that starting exercise approximately 1 h after consuming 15 g gelatin results in greater collagen synthesis during post-exercise recovery.

With proper nutrient availability, exercise optimized collagen synthesis in engineered ligaments [82,98]. Shaw et al. [81] had subjects ingest a vitamin C-enriched gelatin supplement three times/day for 3 days, with ≥6 h between 6 min jump roping sessions. Supplementation increased circulating glycine, proline, hydroxyproline, and hydroxylysine levels, peaking 1 h after ingestion. Engineered ligaments were then treated for 6 days with serum from samples collected before or 1 h after subjects consumed either a placebo supplement, 5 g of gelatin, or 15 g of gelatin. Subjects who ingested gelatin had increased collagen content and better engineered ligament mechanics. Subjects who ingested 15 g gelatin 1 h before exercise doubled the amino-terminal propeptide of collagen I in their blood, indicating better collagen synthesis. Shaw et al. [81] concluded that adding gelatin to an intermittent exercise program improved the collagen synthesis that could facilitate ECM repair and help prevent ligament injury.

Regular essential amino acid consumption in the form of collagenous protein supplements prior to exercise might benefit ACL recovery from training [93]. Within in vitro ligament models, mechanical loading bouts have been found to be sufficient to optimize collagen production [82]. Key nutrient (proline and ascorbic acid) bioavailability combined with the applied strain stimulus produced superior collagen synthesis responses [98]. Additionally, the amino acids glycine and proline increased collagen production and engineered ligament mechanics [98]. Shaw et al. [81] reported enhanced in vitro and in vivo collagen synthesis following ingestion of nutrient-dense foods. Well-designed and timed exercise and nutrition interventions may also support a more rapid post-injury recovery [96].

Optimizing key nutrient bioavailability during ligament collagen synthesis up-regulation from exercise-based mechanical stimulation can better facilitate microtrauma-induced ECM repair and regeneration [96]. Since ACL blood flow is reduced during loading, amino acid bioavailability from protein consumption [96] and supplemental therapeutic agents with proper sleep may take on greater importance to prolong circulatory nutrient delivery to healing tissues. Preclinical studies have shown that bone healing and Type 1 collagen synthesis are enhanced with vitamin C supplementation, while reactive oxidative stress parameters are reduced [99].

## 14. Sleep and the Autonomic Nervous System

The sympathetic nervous system plays a role in the pathology associated with chronic joint inflammation. Dysregulated autonomic nervous system function can lead to tissue destruction from chronic inflammatory disease. Any behavioral or psychological event can influence how the autonomic nervous system influences post-injury tissue healing [100]. Autonomic nervous system organization, in the peripheral nervous system and in the CNS is the basis for cardiovascular system regulation, thermoregulation, fluid balance regulation, evacuative function regulation [pelvic organs], energy balance and nutrition [including the gastrointestinal tract], circadian timing regulation, and body disease protection regulation [including immune system function] [101].

During healthy sleep, parasympathetic nervous system activation is normally greater compared to its activation during wakefulness, whereas sympathetic nervous system activation is reduced [102]. Increased sympathetic nervous system activation during sleep could affect sleep quality and duration leading to decreased well-being and reduced long-term health [103]. Quality sleep helps protect the cardiovascular system by lowering blood pressure levels and resetting baroreceptor sensitivity. Consequently, sympathetic nervous system up-regulation during sleep may counteract its protective functions [104], being inversely related to sleep quality [105].

Healthy sleep benefits multiple biological functions and physiological processes including learning, memory and cognition [106]. Experiencing less than 6 h of sleep nightly for more than three consecutive nights disturbs glucose metabolism, appetite regulation, immune system function, impairs cognition and disturbs mood. Evening cell phone use has been associated with altered sleep–wake cycles and diminished autonomic nervous system activation upon awakening [107]. Dietary supplements that positively influence both sleep duration and sleep quality include protein ingested prior to sleeping. This nutritional intervention can both enhance sleep and improve overnight protein synthesis [106]. They may be especially important for ACL microtrauma recovery and injury prevention.

## 15. Entheses and Crimp

There is a direct relationship between ACL collagen crimp frequency and myofibroblast density [108]. Murray et al. [109] hypothesized that myofibroblasts were involved in ACL crimp formation and tensile force transmission. Crimp is an integral part of ACL function because it influences its viscoelastic mechanical properties during joint loading, providing some initial compliance when it is stretched from a resting state. In a mechanical stress–strain curve, this phenomenon is referred to as the “toe region” [61]. Contemporary biomechanical research does not adequately appreciate regional ACL histological differences such as neurovascular networks, mechanoreceptor density, crimp patterns, or enthesis characteristics, instead inaccurately portraying the ACL as a homogenous structure. Using an ovine model, Zhao et al. [61] demonstrated different ACL bundle crimp morphologies at different knee flexion angles. Based on these crimp pattern differences, the anteromedial bundle was found to function more during weightbearing (stance) with the highest crimp concentration located in its medial portion. The posterolateral bundle was found to be loaded more at maximum knee extension and flexion positions in association with increasing transverse plane tibial rotation [61]. In the presence of reduced crimping under sudden loads, the ACL undergoes a quicker, stiffer response (smaller toe region) when suddenly stretched, potentially increasing its acute injury risk [110,111].

## 16. Primary Prevention: Re-Thinking the Approach

The opportunity to decrease the non-contact, mechanical fatigue or overuse-related ACL injury incidence of youth and adolescent sport athletes is a fertile, and up to now, highly untapped primary injury prevention opportunity. Given that this population is at the greatest risk for sustaining ACL injury, logic suggests that, if possible, we should do more to protect them. In combination with the natural neurovascular ACL constriction that occurs with exercise loading and the dilation that occurs with active recovery (low-impact aerobic activities), respectively, appropriately timed physical therapeutic interventions may accentuate neurovascular and lymph system pump responsiveness to more quickly deliver tissue healing biological fluids and remove metabolic waste products [51]. These modalities in combination with the right exercise dosage, a diet rich in nutrients that facilitate ACL collagen synthesis [57,81,82,94,95,96,97,98], proper sleep patterns (autonomic nervous system regulation) [82,98,99,100], and limited or modified social media use (to decrease excessive sympathetic nervous system activation) [112] may greatly reduce the incidence of these injuries.

In today’s elite-type training model, many youth and adolescent sport athletes train and compete year round in one sport, often participating on more than one team. This sets the stage for ACL ECM degradation and a greater likelihood for sudden fatigue-related failure from accumulated microtrauma, particularly near the highly complex ACL entheses where uncalcified soft tissue sequentially transitions to calcified bone [22]. Sympathetic nervous system dysregulation with prolonged hyper-arousal and elevated plasma cortisol levels at night is associated with the poor sleep patterns that contribute to poor ACL healing from accumulated microtrauma.

Fortunately, prior to experiencing the macro-traumatic ACL “pop” indicative of sudden mechanical fatigue ACL failure, if provided with a proper combination of modified sports training, active recovery time [3], healthy sleep patterns, and proper nutrition [81,93], the native ACL of youth and adolescent athletes has the capacity to completely heal from accumulated microtrauma and even become structurally stronger [27] (Figure 7). To date, sport training periodized to improve the power or endurance of neuromuscular or cardiopulmonary systems has not placed sufficient attention on the natural healing potential of the ACL.

## 17. Conclusions

The unique natural histological heterogeneity, structure and function of the ACL warrants greater attention. A more appropriate balance between sport and exercise training loads with active recovery in combination with proper nutrition and sleep, has the potential to restore ACL collagen balance homeostasis. Enhanced ECM structural remodeling could prevent the accumulated microtrauma that precedes non-contact ACL injury from sudden mechanical fatigue failure. Greater awareness and implementation of recovery science primary prevention factors may decrease the ACL injury incidence of youth and adolescent athletes.

## Figures and Tables

**Figure 1 jfmk-09-00080-f001:**
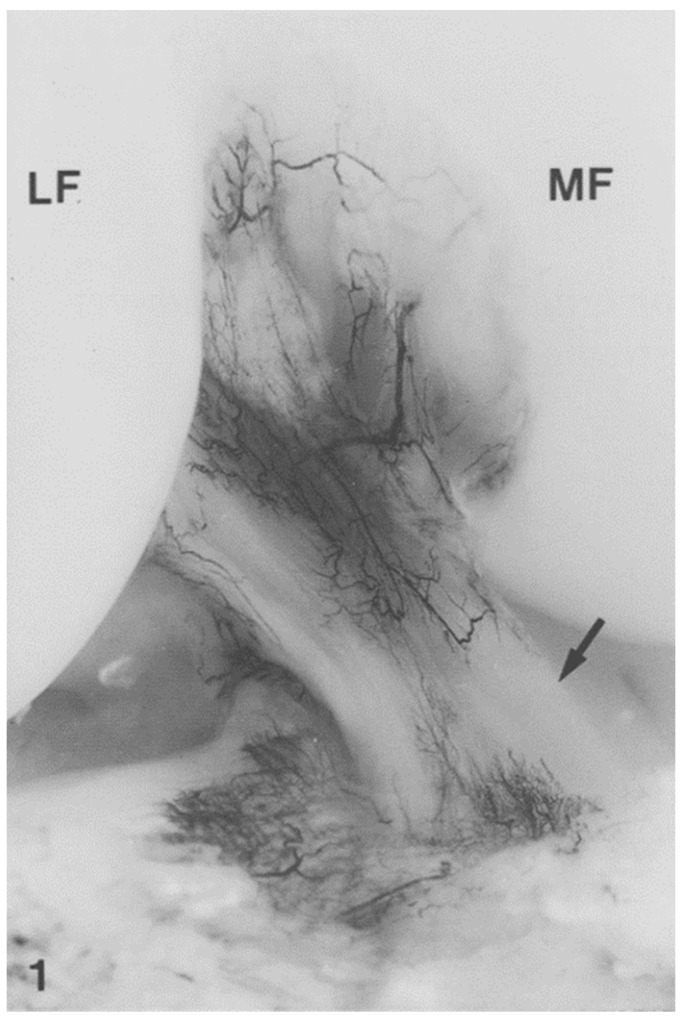
India ink injection. The major blood supply of the anterior cruciate ligament (ACL) arises from the middle geniculate artery. The distal part of the ACL is vascularized by branches of the inferior geniculate artery. The terminal branches of the middle and the inferior geniculate artery form a periligamentous network. In the anterior part of the ACL, 0.5 cm proximal to the tibial insertion where the ligament is in contact with the notch in full knee extension, the periligamentous network of blood vessels is absent (arrow). MF, medial femoral condyle; LF, lateral femoral condyle. From: [51]. Used with the permission of Elsevier Publishers.

**Figure 2 jfmk-09-00080-f002:**
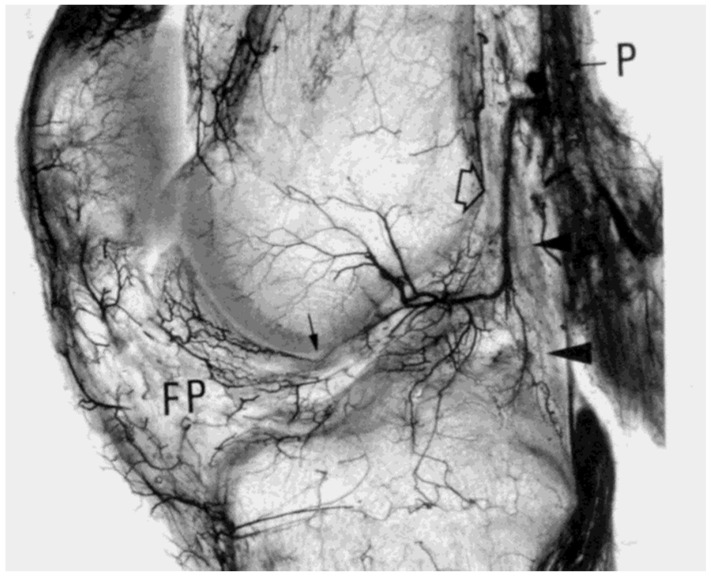
Spalteholz preparation of a 2 cm thick sagittal section of the right knee (33-year-old male), showing the origin at right angles of the middle genicular artery (open arrow) from the popliteal artery (P), its almost vertical crossing of the posterior capsule (arrowheads), and its intraarticular osseous and soft tissue distribution. The descending branches for the cruciate ligaments are clearly visible. The small arrow in front of the ACL indicates some arterioles within the ligamentum mucosum, apparently anastomosing with the intercondylar descending branches of the middle genicular artery, and the infrapatellar fat pad (FP). From: [50]. Used with the permission of John Wiley & Sons Publishing.

**Figure 3 jfmk-09-00080-f003:**
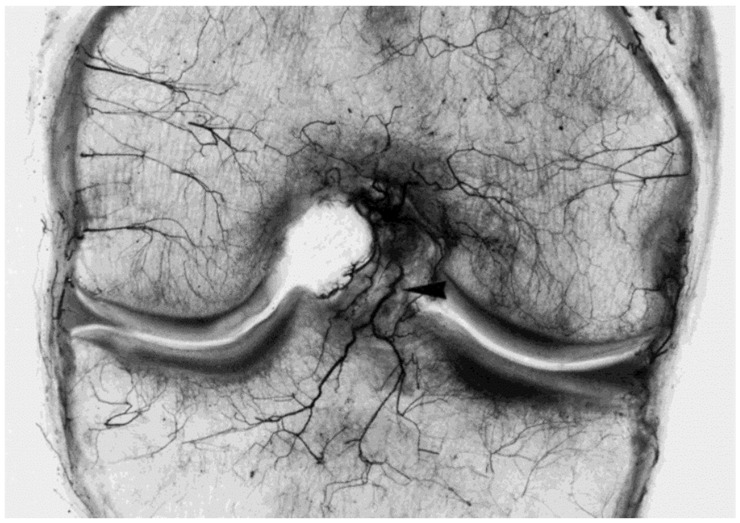
Anterior view of a coronal section about 2 cm thick through the middle part of the left knee (46-year-old male), from which the posterior cruciate ligament was removed. Spalteholz preparation. Besides receiving nutrient arteries, the ACL sustains a large, posteriorly descending branch of the middle genicular artery directed to the upper tibia (arrowhead). The radiate intercondylar and condylar arteries of the femur are also shown. From: [50]. Used with the permission of John Wiley & Sons Publishing.

**Figure 4 jfmk-09-00080-f004:**
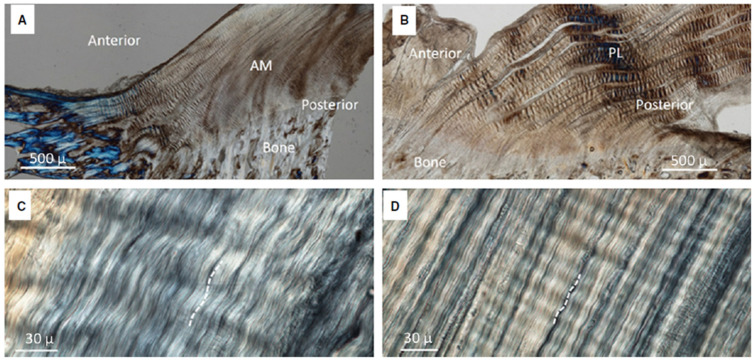
Crimping in (**A**) AM bundle and (**B**) PL bundle. Coarse crimps were visible in the anterior region of the AM bundle and throughout the PL bundle. (**C**) A high magnification view of the coarse crimping in the AM bundle and (**D**) fine crimping of the posterior region of the AM bundle. From: [61]. Used with the permission of John Wiley & Sons Publishing.

**Figure 5 jfmk-09-00080-f005:**
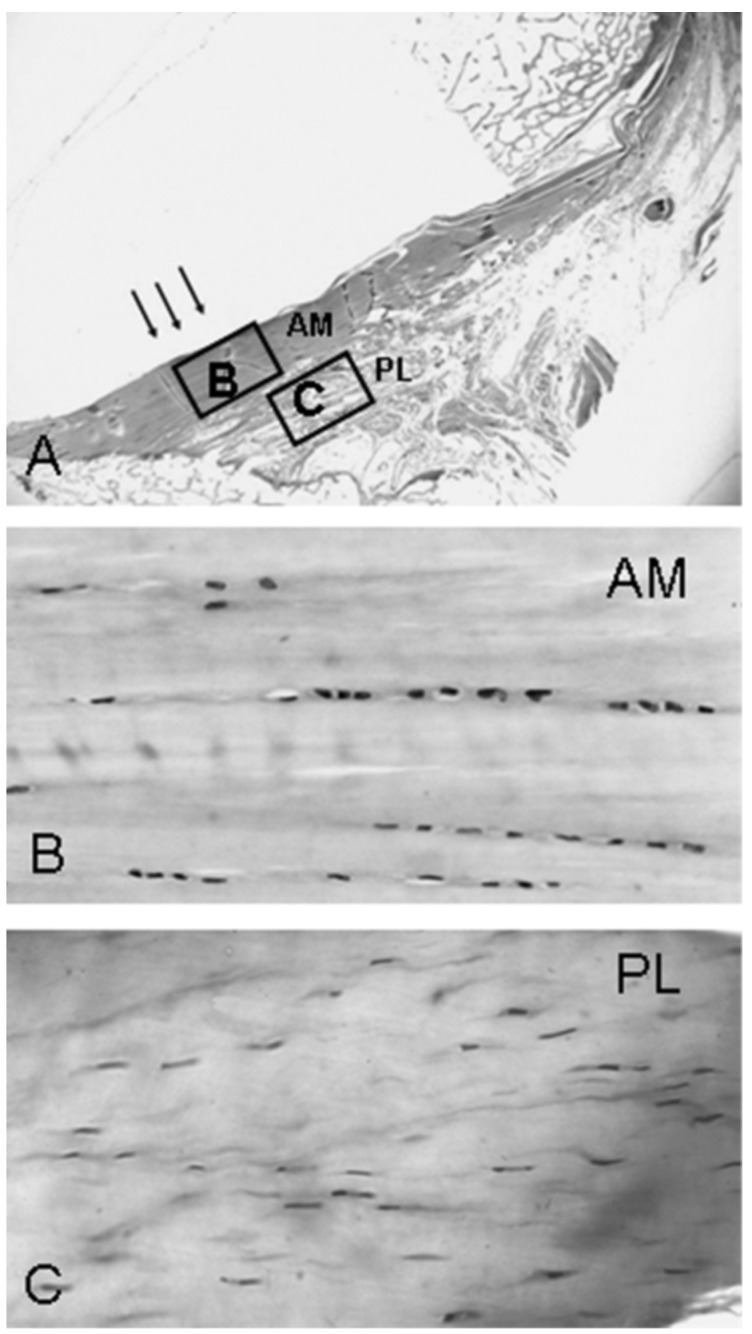
(**A**–**C**) This is a Toluidine blue-stained histological section of the anterior part of the distal third of the ACL (original magnification, ×5). (**B**) In the part where the ACL is in contact with the fossa (“physiological impingement”) (arrows), chondrocyte-like cells aligned in rows can be found (original magnification, ×40). (**C**) In the PL bundle, there are fusiform cells aligned between the collagen fibers (original magnification ×40). From: [52]. Used with the permission of Wolters Kluwer Health, Inc.

**Figure 6 jfmk-09-00080-f006:**
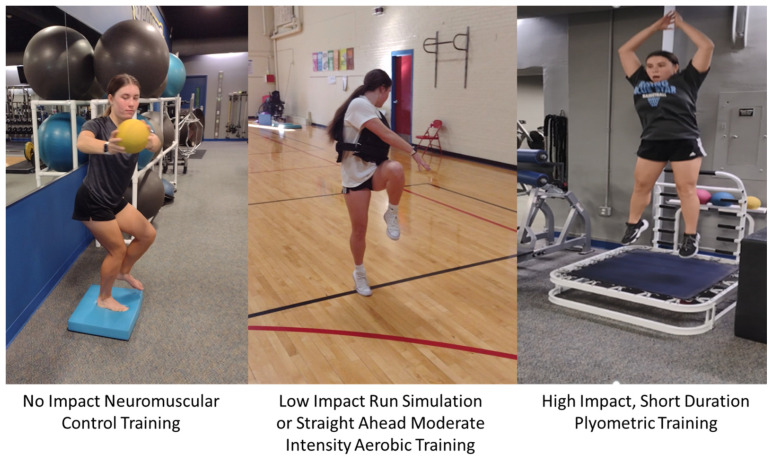
Intermittent active recovery days consisting of neuromuscular control training, progressive intensity aerobic endurance exercises and short plyometric training bouts with adjusted sport training may facilitate the ACL ECM structural and functional improvements that with proper nutrition and sleep may overcome ACL ECM microtrauma accumulation.

**Figure 7 jfmk-09-00080-f007:**
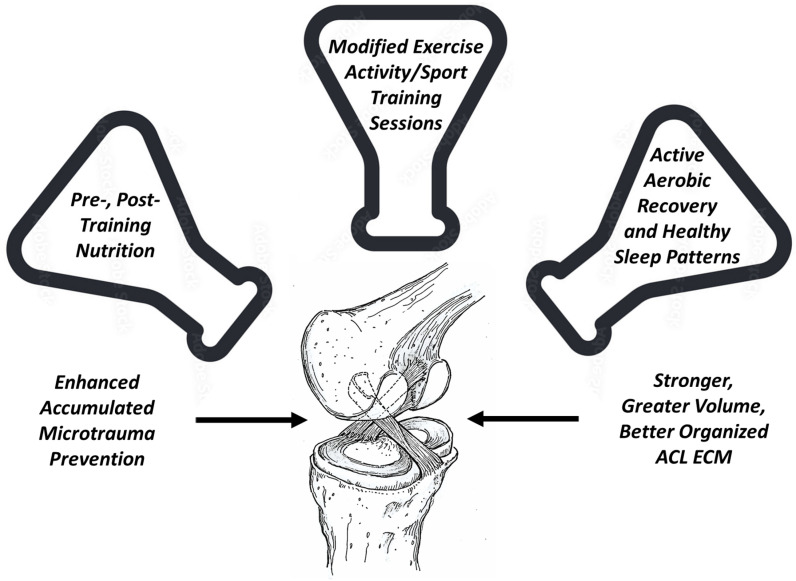
The right combination of pre- and post-training nutrition, modified exercise activity/sport training sessions, and active aerobic recovery with proper sleep can reset sympathetic and parasympathetic nervous system balance to better facilitate ACL microtrauma healing.

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
