# Peer review of "Regenerative Anterior Cruciate Ligament Healing in Youth and Adolescent Athletes: The Emerging Age of Recovery Science"

_jfmk, 2024, doi:10.3390/jfmk9020080_

Round 1
Reviewer 1 Report
Comments and Suggestions for Authors
Please see attached.

Reviewer 2 Report
Comments and Suggestions for Authors
Dear authors,
dear editorial board,
thank you for the opportunity to review this manuscript. Overall, this manuscript is very well written and sufficiently covers all relevant data in the field.
The only concern I have is the manuscript's structure, as it is, in parts, tough to follow. I would suggest to re-arrange it and moving the sections "native ACL blood supply", "Regional AcL HIstological Differenes and Complexities", "The Extracellular Matrix", "Causal Tissue Histogenesis", and "Homeostasis Modulation" to the front after Introduction and Epidemioloy. This may be followed by the "limitations of ACL reconstruction", "surgical ACL repair", "regenerative orthobiologics", and the rest. Alternatively, another structure that the authors feel to be appropriate.
Author Response
Please see our responses to each comment or concern.

Reviewer 3 Report
Comments and Suggestions for Authors
Dear authors, thank you for the opportunity to review your manuscript. The paper is very complex, and full of ideas, but lacks a clear objective. Furthermore, there are several paragraphs, of a didactic-informative nature rather than of an in-depth analysis of the orthobiological impact of ACL rehabilitation.
L36 Colloquialism, among other things in the abstract there is no mention of the high incidence in the female gender..
L42-L49 take the concept of "recovery science" for granted.. it is not clear, and I cannot find in the literature the role of "recovery science" with ACL rehabilitation..
The introduction misses the point, the goal of this review
Are paragraphs 2 and 3 necessary for writing your review? I think I understand that the focus is on the orthobiological role.. I don't recommend removing them, but leaving some mention in the introduction..
In paragraph 4 which acts as a bridge between the introduction and the section of the manuscript there is only one bibliographical reference.
Paragraph 8 seems like a didactic-introductory paragraph in this position what insight would it bring to the reader?
Is paragraph 10 necessary in your review? another manuscript would be needed to describe exercise dosage. Sincerely, recommending removal..
At the same time, paragraph 13, like 14, is sometimes superfluous due to the nature of your paper with respect to the neurobiological impact of the objective
Author Response
Please see our responses to each review comment or concern.

Reviewer 4 Report
Comments and Suggestions for Authors
In general
It is an interesting review, but iThenticate reported that this paper showed 47% wording duplication in the manuscript.
There are too many widespread topics related to ACL injuries, and the content of each topic seems to be somewhat lacking. The authors should focus on a few issues with in-depth content.
3. The Limitations of ACL Reconstruction
The authors should add some reviews or meta-analyses of the clinical outcomes of ACL reconstruction and return to sports activities.
4. Surgical ACL Repair: Growing Evidence
Previous studies showed poor evidence of self-healing of torn ACL. Therefore, ACL reconstruction has been used as the golden standard for injured patients instead of conservative therapy such as cast immobilization. Please show more evidence of the self-healing potential of the torn ACL.
10. Exercise Dosage
Almost all studies concerning the topics are experimental ones. Please show the clinical evidence.
12. Homeostasis Modulation
The authors stated, "Exercise load characteristics can influence both tissue histology and the synovial fluid environment." Is it true in the case of a torn ACL? Please show the evidence.
Conclusion
The authors stated, "Enhanced ECM structural remodeling can prevent the accumulated micro trauma that precedes non-contact ACL injury from sudden mechanical fatigue failure." Is there any clinical evidence for it?
Comments on the Quality of English Language
iThenticate reported that this paper showed 47% wording duplication in the manuscript. Please re-write the sentences.
Reviewer 5 Report
Comments and Suggestions for Authors
First of all, I would like to thank the editorial team for proposing me for this review. I would like to congratulate the authors for their hard work.
Some minor issues that need to be addressed before publishing the article:
- It would be interesting to mention the most common injury mechanism of ACL rupture when it is non-impact (perhaps in sections 1 or 2).
- Since at the end of section 3 they speak of the differences in terms of stability according to the graft used. I would take the opportunity to talk a little earlier about the types of grafts that can be performed, at least the most common ones.
- There are some regions of the article that lack scientific justification (citations). For example the lines 276-288. I recommend reviewing the entire manuscript in this regard and try to provide as much scientific justification as possible.
The manuscript does not follow the structure of a traditional manuscript (introduction, method, results and discussion), but I understand that this is not a problem for the journal.
Again, congratulations to the authors for their work. I find it very interesting and the only thing that worries me is the high percentage of similarity (almost 50%). I leave the decision to the editors on this issue.
Round 2
Reviewer 3 Report
Comments and Suggestions for Authors
Dear Authors, thank you for the opportunity to review the manuscript, unfortunately I fear I must continue to suggest that the manuscript is unsuitable for publication. There are gaps in continuous bibliographic references; there is a peremptory reference to "recovery science", but it has no roots in literature; there is still a lack of a clear objective, the authors end up discussing sections of nutrition, and Sleep and approaches to the Autonomic Nervous System.
Furthermore,
"what is needed to reduce the accumulated microtrauma that leads to sudden, noncontact ACL rupture in many youth and adolescent sport athletes".
I think it's a misleading statement, on the other hand since it's more common in females, does it mean they're more subject to microtraumas?
Author Response
Thanks for helping us improve the manuscript. Our responses to your concerns are addressed in the attached document. Regards, The Authors

Reviewer 4 Report
Comments and Suggestions for Authors
Authors reply: Our responses are in italics. REVIEWER #4 It is an interesting review, but iThenticate reported that this paper showed 47% wording duplication in the manuscript. There are too many widespread topics related to ACL injuries, and the content of each topic seems to be somewhat lacking. The authors should focus on a few issues with in-depth content. • We requested and reviewed the iThenticate document shared by the editorial management team. The largest part of this duplication score appears to be related to the figure legends for which we obtained permission for reproduction in this original paper. In addition to this we have reviewed the manuscript to rephrase certain sections when possible. These changes have been highlighted in blue.
The reviewer's comment: I understand.
• Although many widespread topics have been included in this review commentary, each possesses important considerations related to curtailing the cycle of accumulated microtrauma that precedes non-contact ACL rupture from mechanical fatigue failure.
The reviewer's comment: I understand.
• Where possible we have added key references to support the presented ideas and concepts. New information and reference number changes have been highlighted in green. 3. The Limitations of ACL Reconstruction The authors should add some reviews or meta-analyses of the clinical outcomes of ACL reconstruction and return to sports activities. We have added the systematic review of Ardern et al. (2014) to Ardern et al. (2011): Ardern, C.L.; Webster, K.E.; Taylor, N.F.; Feller; J.A. Return to sport following anterior cruciate ligament reconstruction surgery: A systematic review and meta-analysis of the state of play. Br J Sports Med 2011, 45, 596-606. Ardern, C.L.; Taylor, N.F.; Feller, J.A.; Webster, K.E. Fifty-five per cent return to competitive sport following anterior cruciate ligament reconstruction surgery: An updated systematic review and meta-analysis including aspects of physical functioning and contextual factors. Br J Sports Med 2014; 48, 1543-1552. In addition to the meta-analyses, we have added an editorial that provides an overview of ACL reconstruction deficiencies: Wojtys, E.M. The Missing Link. Sports Health 2023, 15, :9-10 …and made reference to two papers that identify residual neurosensory impairments following ACL rupture and reconstruction: Nyland, J.; Gamble, C.; Franklin, T.; Caborn, D.N.M. Permanent knee sensorimotor system changes following ACL injury and surgery. Knee Surg Sports Traumatol Arthrosc 2017, 25, 1461-1474. 2 Nyland, J.; Wera, J.; Klein, S.; Caborn, D.N. Lower extremity neuromuscular compensations during instrumented single leg hop testing 2-10 years following ACL reconstruction. Knee 2014, 21, 1191-1197. 4. Surgical ACL Repair: Growing Evidence Previous studies showed poor evidence of self-healing of torn ACL. Therefore, ACL reconstruction has been used as the golden standard for injured patients instead of conservative therapy such as cast immobilization. Please show more evidence of the self-healing potential of the torn ACL. We have added this reference related to non-surgical management of a ruptured ACL: Filbay, S.R.; Dowsett, M.; Chaker Jomaa, M.; Rooney, J.; Sabharwal, R.; Lucas, P.; Van Den Heever, A.; Kazaglis, J.; Merlino, J.; Moran, M.; Allwright, M.; Kuah, D.E.K.; Durie, R.; Roger, G.; Cross, M.; Cross, T. Healing of acute anterior cruciate ligament rupture on MRI and outcomes following non-surgical management with the Cross Bracing Protocol. Br J Sports Med 2023 57, 1490-1497. However, please be clear that this review commentary focuses on considerations related to preventing the accumulated microtrauma that leads to sudden, non-contact ACL mechanical fatigue failure. This is a new concept with emerging literature. We have added: Wojtys, E.M.; Beaulieu, M.L.; Ashton-Miller, J.A. New perspectives on ACL injury: On the role of repetitive submaximal knee loading in causing ACL fatigue failure. J Orthop Res 2016, 34, 2059-2068. Wojtys, E.M. The Missing Link. Sports Health 2023, 15, :9-10 This review commentary is not focusing on a partially or completely torn ACL. Rather, we focus on considerations about how primary prevention and how to better prevent sudden, non-contact ACL rupture from the accumulated microtrauma that occurs with high frequency, high intensity, poorly regulated loading. The commentary is conceptual, however, the evidence to date, is appropriately cited. 10. Exercise Dosage Almost all studies concerning the topics are experimental ones. Please show the clinical evidence. This is an emerging topic with more basic science than clinical science evidence at this point. However, we provide the best and latest information. We have added the following basic and clinical research study references: Loflin, B.E.; Ahn, T.; Colglazier, K.A.; Banaszak Holl, M.M.; Ashton-Miller, J.A.; Wojtys, E.M.; Schlecht, S.H. An Adolescent Murine In Vivo Anterior Cruciate Ligament Overuse Injury Model. Am J Sports Med 2023, 51, 1721-1732. Grodman, L.H.; Beaulieu, M.L.; Ashton-Miller, J.A.; Wojtys, E.M. Levels of ACL-straining activities increased in the six months prior to non-contact ACL injury in a retrospective survey: Evidence consistent with ACL fatigue failure. Front Physiol, 2023, 14:1166980. 3 12. Homeostasis Modulation The authors stated, "Exercise load characteristics can influence both tissue histology and the synovial fluid environment." Is it true in the case of a torn ACL? Please show the evidence. Concepts related to partially or completely torn ACL injury management are beyond the scope of this commentary. Rather, we focus on the primary prevention of sudden, non-contact ACL rupture from the accumulated microtrauma that occurs with high frequency, high intensity, poorly regulated loading. These three additional references help to better delineate and support this focus. Wojtys, E.M.; Beaulieu, M.L.; Ashton-Miller, J.A. New perspectives on ACL injury: On the role of repetitive submaximal knee loading in causing ACL fatigue failure. J Orthop Res 2016, 34, 2059-2068. Wojtys, E.M. The Missing Link. Sports Health 2023, 15, :9-10 Grodman, L.H.; Beaulieu, M.L.; Ashton-Miller, J.A.; Wojtys, E.M. Levels of ACL-straining activities increased in the six months prior to non-contact ACL injury in a retrospective survey: Evidence consistent with ACL fatigue failure. Front Physiol, 2023, 14:1166980.
The reviewer's comment: I understand.
Conclusion The authors stated, "Enhanced ECM structural remodeling can prevent the accumulated micro trauma that precedes non-contact ACL injury from sudden mechanical fatigue failure." Is there any clinical evidence for it?
We have added basic science and clinical science evidence to support this. Loflin, B.E.; Ahn, T.; Colglazier, K.A.; Banaszak Holl, M.M.; Ashton-Miller, J.A.; Wojtys, E.M.; Schlecht, S.H. An Adolescent Murine In Vivo Anterior Cruciate Ligament Overuse Injury Model. Am J Sports Med 2023, 51, 1721-1732. Grodman, L.H.; Beaulieu, M.L.; Ashton-Miller, J.A.; Wojtys, E.M. Levels of ACL-straining activities increased in the six months prior to non-contact ACL injury in a retrospective survey: Evidence consistent with ACL fatigue failure. Front Physiol, 2023, 14:1166980. We agree that the basic science evidence is stronger at this point. However, in our opinion, the information that we share, although early in clinical study support has the potential to positively impact the behaviors of everyone involved with adolescent athletes who are at risk for non-contact ACL injury from sudden mechanical fatigue failure. The study by Grodman et al. (2023) is particularly impactful.
The reviewer's comment: I understand.
47% wording duplication, please rewrite the sentences. We have reviewed the manuscript based on the iThenticate report and have applied edits, shown in blue highlighted text.
The reviewer's comment: I understand.
The authors replied to my questions politely.
Still, I am concerned about the conclusion.
"A more appropriate balance between sport and exercise training loads, with active recovery, in combination with proper nutrition and sleep, can restore ACL collagen-balance homeostasis. Enhanced ECM structural remodeling can prevent the accumulated microtrauma that precedes non-contact ACL injury from sudden mechanical fatigue failure. "
The word "can" is too strong an expression, and I would like to change it to "could." or " has the potential to."
Round 3
Reviewer 3 Report
Comments and Suggestions for Authors
-
Author Response
We did not see any further comments or questions from Reviewer #3.